# Analysis of CFTR mRNA and Protein in Peripheral Blood Mononuclear Cells via Quantitative Real-Time PCR and Western Blot

**DOI:** 10.3390/ijms25126367

**Published:** 2024-06-08

**Authors:** Alexander Schnell, Stephanie Tamm, Silke Hedtfeld, Claudio Rodriguez Gonzalez, Andre Hoerning, Nico Lachmann, Frauke Stanke, Anna-Maria Dittrich, Antje Munder

**Affiliations:** 1Department of Pediatric and Adolescent Medicine, University Hospital Erlangen, 91054 Erlangen, Germany; andre.hoerning@uk-erlangen.de; 2Department of Pediatric Pneumology, Allergology and Neonatology, Hannover Medical School, 30625 Hannover, Germany; 3Biomedical Research in Endstage and Obstructive Lung Disease Hannover (BREATH), German Center for Lung Research, Hannover Medical School, 30625 Hannover, Germany; 4Fraunhofer Institute for Toxicology and Experimental Medicine, 30625 Hannover, Germany

**Keywords:** CFTR, cystic fibrosis, expression, protein, RNA, Western blot, qPCR, immune cells, leukocytes, immune cell lines

## Abstract

The *Cystic Fibrosis Conductance Transmembrane Regulator* gene encodes for the CFTR ion channel, which is responsible for the transport of chloride and bicarbonate across the plasma membrane. Mutations in the gene result in impaired ion transport, subsequently leading to perturbed secretion in all exocrine glands and, therefore, the multi-organ disease cystic fibrosis (CF). In recent years, several studies have reported on CFTR expression in immune cells as demonstrated by immunofluorescence, flow cytometry, and immunoblotting. However, these data are mainly restricted to single-cell populations and show significant variation depending on the methodology used. Here, we investigated *CFTR* transcription and protein expression using standardized protocols in a comprehensive panel of immune cells. Methods: We applied a high-resolution Western blot protocol using a combination of highly specific monoclonal CFTR antibodies that have been optimized for the detection of CFTR in epithelial cells and healthy primary immune cell subpopulations sorted by flow cytometry and used immortalized cell lines as controls. The specificity of CFTR protein detection was controlled by peptide competition and enzymatic Peptide-N-Glycosidase-F (PNGase) digest. *CFTR* transcripts were analyzed using quantitative real-time PCR and normalized to the level of epithelial T84 cells as a reference. Results: *CFTR* mRNA expression could be shown for primary CD4^+^ T cells, NK cells, as well as differentiated THP-1 and Jurkat T cells. In contrast, we failed to detect *CFTR* transcripts for CD14^+^ monocytes and undifferentiated THP-1 cells, as well as for B cells and CD8^+^ T cells. Prominent immunoreactive bands were detectable by immunoblotting with the combination of four CFTR antibodies targeting different epitopes of the CFTR protein. However, in biosamples of non-epithelial origin, these CFTR-like protein bands could be unmasked as false positives through peptide competition or PNGase digest, meaning that the observed mRNA transcripts were not necessarily translated into CFTR proteins, which could be detected via immunoblotting. Our results confirm that mRNA expression in immune cells is many times lower than in that cells of epithelial origin. The immunoreactive signals in immune cells turned out to be false positives, and may be provoked by the presence of a high-affinity protein with a similar epitope. Non-specific binding (e.g., Fab-interaction with glycosyl branches) might also contribute to false positive signals. Our findings highlight the necessity of accurate controls, such as CFTR-negative cells, as well as peptide competition and glycolytic digest in order to identify genuine CFTR protein by immunoblotting. Our data suggest, furthermore, that CFTR protein expression data from techniques such as histology, for which the absence of a molecular weight or other independent control prevents the unmasking of false positive immunoreactive signals, must be interpreted carefully as well.

## 1. Introduction

The *Cystic Fibrosis Conductance Transmembrane Regulator* (CFTR) gene encodes for the CFTR protein, functioning as a transmembrane ion channel transporting chloride and bicarbonate across the plasma membrane. Mutations in the gene result in impaired ion transport, subsequently leading to perturbed secretion of chloride and bicarbonate by all exocrine glands, creating a hyperviscous product, which causes the multi-organ disease cystic fibrosis (CF) [1]. Since its discovery in 1989 by Riordan and coworkers, CFTR has been described as mainly expressed in cells of epithelial origin. However, CFTR mRNA expression has also been shown in non-epithelial cells by Yoshimura et al. [2]. Electrophysiological experiments suggest a functional role of CFTR in immune cells as well [3,4]. In the context of CF, defective or overshooting immune cell responses of several myeloid immune cells, like monocytes/macrophages [5,6,7] or neutrophils [8], as well as a pathogenic involvement of adaptive immune cells, have been described [9,10].

In the past decades, improvements in CF therapy have increased the median age of survival from 4 to 5 years in the 1950s to 50 years in Europe today [11]. Until the last decade, the medical regimen was limited to symptomatic treatment. However, the development of a new class of drugs, namely, CFTR modulators, has set a new milestone in the treatment of CF, as these drugs directly target the malfunctioning CFTR protein, providing a mutation-specific medical treatment which can now be offered to a majority of all patients. Since the first market admission of such drugs in 2012, a number of reports have shown that CFTR modulators also affect immune cell function [12,13,14,15]. In this light, the question arises as to whether these reported effects are due to a direct drug–CFTR interaction in immune cells or, rather, whether they are caused by secondary, by-proxy effects. Therefore, the concise evidence of a relevant CFTR protein expression becomes of critical importance. To date, several publications have suggested CFTR protein expression by immune cells, demonstrated by Western blot or flow cytometry [16]. However, these data are mainly restricted to single cell populations and show significant variation depending on the methodology used. Moreover, some of the data remain unconcise, since they lack controls to prove the specificity of the depicted signals.

In the study presented herein, we include a comprehensive panel of immune cells, consisting of primary immune cells directly isolated from healthy volunteers and immortalized cell lines. In experimental settings, the use of immortalized cell lines is common, since these cells are easily available; they can be cultured in a highly standardized manner; and the interindividual variability, which is often seen in patient-derived primary immune cells, can be eliminated. However, immortalized cells are quite far away from the in vivo situation, as they can acquire de novo mutations; do not necessarily reflect specific mutations acquired in vivo; change their epigenetic profile, activation, and differentiation status due to changes in environmental signals encountered in cell culture vs. the natural environment; and might, therefore, no longer be representative of the organ they had originally been isolated from. Furthermore, in some cases, the gene of interest might only be expressed by genetic engineering [17], similarly not necessarily representing the in vivo situation. Primary cells reflect the in vivo situation more closely with respect to mutation and activation status, as well as genetic and epigenetic make-up. However, they are available only in restricted amounts in terms of their number and viability and might have altered their activation status due to isolation procedures; therefore, general statements are allowed only to a limited extent. Thus, each of the cell types used has certain deficiencies in research on CFTR transcription and protein expression.

Here, we determine CFTR transcription and protein expression in primary immune cell subpopulations by focusing on standardized protocols and data analysis for CFTR expression analysis. Thus, we can provide a comprehensive overview that can guide future research into the effects of CFTR-modifying therapeutic approaches.

## 2. Materials and Methods

### 2.1. Cell Culture

#### 2.1.1. Immortalized Cells

##### 16HBE14o-

The respiratory epithelial cell line 16HBE14o- was grown on fibronectin (354008, Corning, Corning, NY, USA) and collagen I (354231, Corning)-coated plates (83.3900.002, Sarstedt, Nümbrecht, Germany) with minimal essential medium (MEM; 12360–038, Thermo Fisher Scientific). It was supplemented with 10% FCS and 1% of 100× penicillin–streptavidin solution (14150–122, Thermo Fisher Scientific) and 1% of 200 mm L-Glutamine solution (258030–081, Thermo Fisher Scientific).

##### T84-

The intestinal cancer epithelial cell line T84 was grown in Dulbecco’s modified Eagle’s medium (DMEM/F12, Thermo Fisher Scientific, Waltham, MA, USA) supplemented with 10% fetal calf serum (FCS) and 1% 100× penicillin–streptavidin–glutamine solution (Thermo Fisher Scientific).

##### THP1

Cells of the monocyte cell line THP1 were cultured in RPMI 1640 medium (21870-076, Thermo Fisher Scientific) supplemented with 10% FCS (S0615, Sigma-Aldrich) and 1% of 100× penicillin–streptavidin solution (14150–122, Thermo Fisher Scientific). THP1 cells were differentiated into macrophage-like cells using 5 ng/mL phorbol 12-myristate 13-acetate (PMA, P8139, Sigma-Aldrich, Taufkirchen, Germany) for 3 d and then used for analyses.

##### HEK-293T Cells

Human embryonic kidney 293T (ATCC^®^ CRL3216TM, LGC Standards, Wesel, Germany) cells were cultured in Dulbecco’s modified Eagle’s medium F12 (DMEM F12, 21331-020 Thermo Fisher Scientific) with 10% FCS and 1% of 100× penicillin–streptavidin solution (14150–122, Thermo Fisher Scientific).

##### Jurkat

Jurkat T cells were obtained courtesy of Prof. Hans-Martin Jäck (Division of Immunology, University Hospital Erlangen, Germany), and were cultured in RPMI containing 10% FCS, 1% penicillin–streptomycin, 1 mM Na-Pyruvat, and 1% ß-Mercapto-EtOH.

### 2.2. Isolation of Primary Immune Cells

PBMCs of two healthy, anonymous volunteers were obtained from the Department of Transfusion Medicine and Hemostasis (University Hospital Erlangen, Germany) as a byproduct of the generation of erythrocyte concentrates. The experiment with PBMCs derived from patients and probands was approved by the local ethics committee (vote # 20-485_1-B). Cells were stored in liquid nitrogen. After thawing, PBMCs were incubated with an antibody cocktail containing CD3, CD4, CD8, CD19, CD14, CD16, CD45, and CD56 for 15 min at room temperature in the dark, and were subsequently washed with PBS containing 1% FCS for 5 min at 300 rpm. Cells were then sorted using a FACS Aria II (Beckton and Dickinson, Franklin Lakes, NJ, USA) into CD4^+^ or CD8^+^ T cells, CD19^+^ B cells, CD14^+^ monocytes, and CD16^+^ CD56^+^ NK cells with a purity of >98%. Isolated immune cell subpopulations were washed √and kept as dry pellets at −80 °C.

#### 2.2.1. Protocol for Immunoblotting and qPCR

mRNA and protein for CFTR were derived from the same biomaterial sample using a branched protocol. For this purpose, whole-cell lysates were prepared from frozen cells in a range of 10^6^–10^7^ and stored at −20 °C. To prepare the lysates, cells were placed into Laemmli buffer with CaCl_2_ (0.5 mmol, 5239.1, Carl Roth, Karlsruhe Germany), 10% glycerol, DTT (0.1 mol, D0632, Sigma Aldrich, Taufkirchen, Germany), protease inhibitor (PI SRE0055, Sigma Aldrich), PMSF at a final concentration of 0.5 mM (P7626, Sigma Aldrich), RNA-free DNAse (M0303, New England Biolabs, Ipswich, MA, USA), and recombinant RNAsin (N252B, Promega, Fitchburg, WI, USA) added. At this point, 20 µL aliquots were frozen for qPCR at −20 °C. The remaining samples were used for immunoblotting.

#### 2.2.2. CFTR Western Blot

For immunoblotting, SDS (final concentration 2%, CN30.3, Carl Roth) and Omnicleave (10 U/µL, OC7850K, Biozym, Hessisch Oldendorf, Germany) were added to cell lysates. Next, the lysates were incubated at 37 °C for 30 min and sheared by pipetting the 50 µL volume ten times with a 200 µL pipette-tip. Centrifugation for 10 min at 13,000 rpm (5424R, Eppendorf, Hamburg, Germany) yielded a supernatant of 35–45 µL, which was adjusted with the same volume of glycerol. The protein content of the lysates was semi-quantified with minute aliquots of 1:5; 1:10; 1:30; and 1:60 serially diluted in 150 mM NaCl. Volumes of 1 µL of the diluted samples were spotted on Whatman 3 MM paper in comparison to a serial dilution of 5.0 μg/μL to 0.1 μg/μL bovine serum albumin in 150 mM NaCl. Spots were dried and stained in Coomassie solution (0.1% Coomassie brilliant blue in 25% isopropanol, 10% acetic acid) for 10 s, and the stained Whatman paper was thoroughly rinsed using running tap water. The protein concentration of the lysates was estimated by comparing the staining of the spotted samples to the staining of the control proteins.

Electrophoresis was carried out in a Mini-PROTEAN Tetra Cell (165-8001; Bio-Rad Laboratories GmbH; Munich, Germany) using 6% polyacrylamide (PAA; Rotiphorese Gel 30, crosslink 37.5:1; Roth; Karlsruhe, Germany). The separation matrix of 6% PAA was cast to yield a separation distance of 6.5 cm below a very narrow 4% PAA gel. Sample volumes were adjusted with 50 mM Tris pH 6.8, 2% (*w*/*v*) SDS, and 50% (*w*/*v*) glycerol, and 100 mM DTT and 2 µL bromophenol blue (115-39-9, Merck, Darmstadt, Germany were freshly added prior to a mild denaturation step of 30 min at 37 °C. 16HBE14o- was used as a positive control and HEK-293T cell lysates as a negative control on each gel. The electrophoretic mobility of the samples was judged against a prestained molecular weight marker (PageRuler Plus Prestained Protein Ladder; 26619; Thermo Fisher; Darmstadt; Germany). Electrophoresis was carried out at 12 V for approximately 20 h at 4 °C, at which point the electrophoresis was continued at 60 V for approximately 3 h until the 72 kDa marker had almost reached the lower edge of the polyacrylamide gel. These conditions had been optimized in preliminary experiments to increase the sensitivity of the immune-chemical CFTR signal, as the optimal resolution was provided within the range of 100–300 kDa.

Proteins were transferred to an uncharged supported nitrocellulose membrane (Amersham Protran Supported Nitrocellulose Blotting-Membrane; 0.45 µm pore size; 10600016; VWR; Darmstadt, Germany) by tank blotting in a Mini Trans-Blot Electrophoretic Transfer Cell (170-3935; Bio-Rad Laboratories GmbH; Munich, Germany). Polyacrylamide gels were mounted into the gel holder cassette, and the high-molecular-weight edge of the gel was placed at the cassette’s hinge. The transfer was carried out in 125 mM Tris, 950 mM glycine, and 0.02% (*w*/*v*) SDS at 44 mA for approximately 23 h, during which the tank blot apparatus was submerged in ice in a Styrofoam container. Upon the completion of the tank blot, the polyacrylamide gel was stained with Coomassie to visualize non-transferred high-molecular-weight proteins. StartingBlock Buffer (Thermo Fisher Scientific) with Tween20 (0.1%) was used for blocking for 1 h at room temperature. The Western blot was then probed with a mixture of four monoclonal antibodies (1:1:1:1 and in a total dilution of 1:1600), or with each antibody, as shown in Appendix A (dilution: 1:400 each), overnight at 4 °C. These antibodies are known to detect different CFTR epitopes; are characterized by high affinity and specificity; and were provided by T. Jensen, Chapel Hill via the antibody distribution program hosted by the CF foundation (see Table 1). Incubation with secondary antibody IgG H&L (HRP) goat anti-mouse (1:7500, ab97040, Abcam, Cambridge, UK) was performed at RT for 1 h. Depending on the signal intensity, two different HRP substrate solutions (SuperSignal West Pico 34078 or SuperSignal West Femto, 34096 Thermo Fisher Scientific Darmstadt, Germany) were applied onto the membranes. Vinculin (MW 117 kDa) was used as a loading control.

To ensure that the signals in all lanes were visualized, the exposure times were varied between 3 s and 30 min for PICO and between 3 s and 10 min for FEMTO, yielding about 15 different exposures of each primary antibody target. Scans were acquired on a DNR-MF-ChemiBIS 3.2 Bio-Imaging System (Berthold Technologies, Bad Wildbad, Germany).

#### 2.2.3. Glycolytic Digest of CFTR Protein

For deglycosylation, 50 µg samples were incubated with 1500 U PNGase F (500 U/µL, P0704L, New England Biolabs) at 37 °C for 3 h prior to Western blotting, as described above.

#### 2.2.4. Peptide Competition

Prior to immunoblotting, 1.5 µL of antibody ab596 (final concentration: 1:400) was pre-incubated with an excessive amount (600 µg) of the peptide H-WPSGGQMT-OH (20 mg/mL PBS, Eurogentech, Köln, Germany) for 1 h at RT. This mixture was placed onto the membrane, and the protocol for immunoblotting was followed as normal (incubation overnight at 4 °C).

#### 2.2.5. qPCR

RNA was extracted from the cell lysates using a Quiagen RNeasy Mini Kit (74104, Qiagen, Hilden, Germany) and a RNase-free DNase Set (79254, Qiagen). Transcription of RNA into cDNA was carried out using a High-Capacity cDNA Reverse Transcription Kit with RNase Inhibitor (4368813, Thermo Fisher Scientific). To detect CFTR RNA, the specific TaqMan probe Hs00357004_m1 came into use with the TaqMan Fast Advanced mastermix (4444556). As a housekeeping gene, GAPDH was identified using the probe Hs02786624_g1 (both from Thermo Fisher Scientific). All samples had been measured in duplicate in the 7500 Fast Real-Time PCR System (Applied Biosystems, Darmstadt, Germany), and the Ct values of CFTR normalized to GAPDH values (2^−ΔΔCt^ method). Samples with Ct values of ≥25 were deemed as unreliable and therefore excluded from analysis. CFTR expression was compared with the epithelial cell line T84 as a reference.

## 3. Results

### 3.1. Quantitative PCR Reveals CFTR Transcripts in Differentiated Immune Cells Lines and Primary Immune Cells

We performed qPCR for immune cell subsets. Each cell sample was split prior to the experiments in order to perform Western blotting and, later on, qPCR from the same biomaterial simultaneously, thus providing truly paired samples (Table 2). For qPCR, bronchial 16HBE and intestinal T84 cells were used. The latter were also used as positive controls to replicate the data provided by McDonald et al. [3] using a commercially available TaqMan kit that relies on specific primers as well as a specific internal probe to detect CFTR mRNA. qPCR was performed on primary immune cells from healthy volunteers, immortalized Jurkat T cells, and differentiated and undifferentiated THP-1 cells. CFTR transcripts were detectable for primary CD4^+^ T cells and NK cells, differentiated THP1 cells, and Jurkat T cells. The expression levels were considerably low, ranging from 100-fold (Jurkat T cells) to 2000-fold (CD16+ NK cells) lower than T84 cells. For undifferentiated THP-1 cells, monocytes, B cells, and CD8^+^ T cells, no CFTR transcripts were detected at all.

### 3.2. Immunoblotting

To confirm the transcript data for protein expression, we resorted to an optimized Western blot protocol using a combination of four CFTR-specific antibodies [18], which enabled us to detect signals from different epitopes across the CFTR protein. When examining CFTR protein expression using Western blotting, we applied the same cells as outlined above for qPCR analyses, including the epithelial cell line 16HBE14o- as a positive control and HEK-293 cells as a negative control. As expected, 16HBE14o- expressed a CFTR signal with a polydisperse C-band at the height of ~160 kDa, representing the complex glycosylated mature protein and, additionally, a distinct, more focused B-band at ~130 kDa (core-glycosylated immature protein). The negative control HEK-293 did not show any CFTR expression (Figure 1).

For the majority of primary immune cells, only a single band of ~130 kDa was visible (green arrows). Notably, for monocytes, only a very faint signal of 130 kDa appeared, accompanied by a band at the height of 95 kDa, which was also detectable for CD4^+^ T cells (red arrows).

From the immortalized cells, the undifferentiated THP1 cells (monocyte-like phenotype) gave no signal at all, whereas the differentiated THP1 cells, representing a macrophage-like phenotype, showed a strong signal of 130 kDa (Figure 1, Appendix A). However, a similar band could be detected in CFTR-negative HEK cells, indicating that this signal did not represent the CFTR protein. Jurkat T cells showed the same pattern as the differentiated THP1 cells (Figure 2). Furthermore, in 16HBE, differentiated and undifferentiated THP1- as well as CFTR-negative HEK control cells, a band with a size of ~220 kDa appeared (Figure 1). It was considered to be non-specific because even the mature CFTR protein appeared with a C-band of 160 kDa size at maximum. It is likely that this band represents a non-CFTR protein that can be detected by CFTR antibodies due to similar epitopes, as was recently described by Sato et al. [20].

When performing the immunoblots with each of the used antibodies separately, we were able to confirm that all of the antibodies gave rise to a CFTR-specific C-band in CFTR-expressing cells (16HBE and T84), whereas no relevant signals could be observed for differentiated THP1 cells nor for HEK cells. However, in those latter two, Ab660 only raised a strong signal at 130 kDa (Appendix A). Hence, it is likely that the presence of Ab660 in the antibody mix that was initially used accounted for the band at 130 kDa which was seen in the immune cell lysates shown in Figure 1.

### 3.3. Deglycosylation of Mature CFTR Protein Detected via Immunoblotting Unveils Unspecific Protein Detection in Immune Cells

In order to further evaluate the specificity of the CFTR signals of primary immune cells and THP1 macrophages and monocytes (Figure 1), enzymatic digest with PNGase, which removes N-linked oligosaccharides from glycoproteins, was performed, again using the mix of four antibodies. Again, 16HBE14o- cells were used as positive controls. As expected in 16HBE14o- cells, after the enzymatic digest with PNGase, the characteristic complex-glycosylated C-bands of the CFTR protein shifted towards the core-glycosylated B-band and, additionally, below 130 kDa, representing the unglycosylated protein also described as band A (Figure 2, blue arrow).

Inconsistently, for CD4^+^ T-cells, differentiated THP-1 cells, and Jurkat T cells, the Western blot bands remained at ~130 kDa and did not shift at all after PNGase treatment (Figure 2, Appendix A). This demonstrates that the protein(s) detected in these immune cells at 130 kDa cannot be deglycosylated with PNGase. Therefore, this suggests that this protein does not represent the core-glycosylated form of CFTR, in spite of the size equivalence at 130 kDa. In contrast, the band of a size of 95 kDa that was observed in the monocytes and T cells (Figure 1) resolved after glycolytic digestion in T cells (Figure 2, Appendix A, red arrow).

### 3.4. Signals Detected by Immunoblotting in Immune Cells Fail to Be Validated by Peptide Competition

To further corroborate that the CFTR protein expression data in immune cells resulted from unspecific antibody binding of other proteins, which did not correspond to CFTR, we performed peptide competition assays. To this end, Western blotting was performed in duplicate, yielding two membranes. One is processed with the CFTR antibody preincubated with the peptide, while the sister membrane was detected in the absence of the competing peptide. Thus, the signals still seen on the former (presence of competing peptide) denoted that the antibody bound unspecifically, while signals detected on the latter (absence of competing peptide) revealed that CFTR was recognized by the antibody’s true epitope.

The CFTR-specificity of the Western blot signal in 16HBE14o-cells was validated by peptide competition, as in the CFTR-C and CFTR-B, signals were only observed in the absence, but not in the presence, of the competing peptide (Figure 3, Appendix A).

On the contrary, signals derived from CD4^+^ and Jurkat T cells THP1 cells, as well as HEK cells, showed the same appearance, irrespective of whether or not a competing peptide was provided to the antibody (Figure 3, Appendix A). Thus, all immunoreactive signals from these cell populations constituted false positive signals corresponding to unspecific bands, corroborating the PNGase experiments.

## 4. Discussion

*CFTR* mRNA expression can be shown for primary, immortalized, and stem cell-derived immune cells via qPCR. However, no *CFTR* transcripts were detectable for CD14^+^ monocytes, undifferentiated THP-1 cells, or B cells, nor for CD8^+^ T cells. A combination of four highly specific monoclonal CFTR antibodies used for immunoblotting detected prominent immunoreactive bands. However, in biosamples of non-epithelial origin, these CFTR-like protein bands could be unmasked as false positive by separate blotting of the antibodies, peptide competition, and PNGase digest. Detection of CFTR from non-epithelial cells was described shortly after the CF disease-causing gene was uncovered [2]. To accomplish that, signals for CFTR mRNA had to be enhanced using a combination of PCR, subsequent blotting of the amplified products, and detection via a CFTR probe labeled radioactively with 32P. These early data suggested that the CFTR mRNA is present in cells beyond epithelial cells, albeit in substantially lower amounts. We also demonstrated CFTR transcript expression in different primary immune cells (CD4^+^ T cells, NK cells) and cell lines (differentiated THP1 cells, Jurkat T cells) using a modern qPCR assay. However, as is similar to what has been reported before in most cell types, mRNA signals by qPCR were at the limit of detection, making quantitative estimates unreliable. Tentatively, we can confirm the expression in those cells to be 100-fold to 2000-fold less than in T84 cells, which is in agreement with the data described by Yoshimura et al. [2].

Next, we tried to confirm the translation of *CFTR* mRNA into CFTR protein using antibodies raised against CFTR. For this, we employed the four monoclonal CFTR antibodies 270, 570, 596, and 660 in combination, using them in a sensitive protocol optimized to characterize CFTR glycoisoforms in patients’ rectal suction biopsies, as described previously [18]. Since these monoclonal antibodies are directed against different epitopes of the complex CFTR protein, immunoblotting as described here is highly specific. However, even this combined specificity cannot prevent the detection of other cellular proteins that contain similar 3D protein structures to those conformations adopted by the CFTR-peptides used to generate the monoclonal antibodies, as we demonstrated for Ab660. In this line, a recent publication identified an amino acid sequence in the ciliary protein rootletin X1 to be similar to a sequence of NBD2, which is detected by Ab596 [20]. A similar mechanism is likely to occur for Ab660 in immune cells as well as in HEK cells.

In our study, we could not confirm mature glycosylated CFTR-C in any of the immune cell-derived SDS-rich whole-cell lysates. Moreover, while we observed an immunoreactive band consistent in size with core-glycosylated CFTR-B, we unmasked this signal as not being CFTR, as it could be neither deglycosylated using PNGase nor blocked by a competing peptide derived from the CFTR sequence.

As initially described, CFTR transcripts have already been described in the early days of CFTR research for almost all primary immune cells present in the peripheral blood (except NK cells); however, there have been considerable efforts to even detect the significantly lower expression level in comparison to epithelial cells [2,3]. Consequently, the question of CFTR protein expression in immune cells has been raised and attempted to be answered in several publications by means of Western blotting; immunohistochemistry or immunofluorescence; and flow cytometry in primary monocytes [16,19], neutrophils [21,22], and T cells or lymphocytes [22,23].

Functional data on the relevance of CFTR in immune cells are mostly derived from animal CFTR knock-out models; usage of patient-derived peripheral blood; or bronchoalveolar innate immune cells like neutrophils [21,24], monocyte-derived cells [7,24,25], or alveolar macrophages [24,25,26]. The research has focused on phagocytosis, production of reactive oxygen species, and secretion of inflammatory cytokines. Additionally, Fan et al. demonstrated a direct relation between a CFTR defect in monocytes and a clinical phenotype by using wild-type and CF mixed bone marrow chimeras, highlighting the functional relevance of CFTR in these cells [27]. From an electrophysiological perspective, CFTR-specific chloride currents in freshly isolated monocytes from healthy volunteers and patients with CF have also been described in patch-clamp experiments [4]. With regard to CFTR expression in adaptive immune cells, the literature is rather scarce; however, there are also data indicating a functional relevance of the secretory profile of adaptive immune cells [10,23,28,29]. This debate has been fueled by the introduction of highly effective modulator therapy. In this area, researchers have demonstrated impressive changes in the distribution and function of immune cells throughout the therapy course [14,30,31,32]. However, although these data prove major improvements regarding the immune systems, caution is warranted in interpretating these results. Especially with regard to our data, which confirm CFTR expression at low levels in some immune cells, potential off-target effects apart from CFTR might be taken into consideration [33].

The interpretation of the existing data on protein expression warrants some caution in terms of the specificity of the materials utilized (especially when polyclonal antibodies are used) and the usage of positive and negative control samples. Taken together, in the absence of further technical confirmation such as peptide competition and/or PNgase digestion, we encourage critical questioning of strong signals raised by antibodies directed against CFTR. In our hands, such signals were regularly unmasked by PNgase digestion or peptide competition as false-positive immunoreactive signals not corresponding to the CFTR protein. Apart from the monoclonal CFTR antibodies provided by Tim Jensen, Chapel Hill via the antibody distribution program for this work [34], we also tested a collection of 10 customized rabbit polyclonal antibodies (Eurogentec) generated to detect CFTR [35,36,37] via immunoblotting. With three of these customized antibodies, we detected a lege artis CFTR signal from 16HBE14o- positive control lysates. However, the signals detected by these antibodies in THP1 and HEK cells—the latter of which are recognized as CFTR-negative—showed the same band patterns as shown in this work, further confirming that THP1 cells do not express CFTR protein in detectable amounts.

Furthermore, our results show that protein expression signals obtained by flow cytometry or immunohistochemistry should be evaluated carefully, as these methods do not permit the application of appropriate quality control measures such as PNGase digestion or peptide competition.

Moreover, the biomaterials compared for the aforementioned studies that used patient-derived samples were obtained from different individuals with diverging genetic backgrounds and CFTR mutations. Thus, the inherited capabilities of the investigated cells might differ with regard to more than only CFTR expression.

In conclusion, while we were able to detect *CFTR* mRNA transcripts in several immune cells via qPCR, the CFTR protein could not be detected by Western blotting in our study, indicating that the ion channel was expressed in very low amounts, if at all, in the analyzed immune cell populations. Due to the high number of studies providing functional evidence of CFTR in immune cells, further studies, e.g., using advanced more precise techniques such as a digital PCR, will be needed to validate CFTR expression in these cells or to unravel the mechanisms that might affect proper *CFTR* mRNA translation. Furthermore, blocking of CFTR-specific antibodies in immunohistochemistry by using competitive peptides could also identify true CFTR signals. Our data highlight the importance of using proper positive and negative controls, and to carry out validation experiments such as glycolytic digestion and peptide competition.

## Figures and Tables

**Figure 1 ijms-25-06367-f001:**
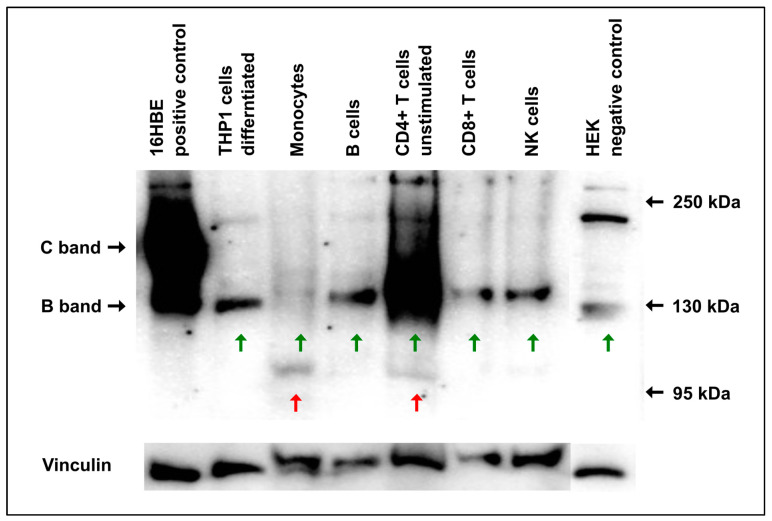
CFTR Western blot of primary immune cells and cells of immortalized cell lines. Primary immune cells (monocytes, NK cells, B cells, CD4^+^ and CD8^+^ T-cells) isolated from the peripheral blood of a healthy subject were compared in terms of their CFTR expression with undifferentiated, monocyte-like THP1 cells and differentiated, macrophage-like THP1 cells. 16HBE14o- cells served as a positive and HEK-293 cells as a negative control, respectively. Both the core-glycosylated CFTR glycoisoform (CFTR-B) and the complex glycosylated mature isoform (CFTR-C) were strongly seen in 16HBE14o- cells. All immune cells, besides undifferentiated THP1 cells, displayed bands compatible with CFTR-B in size (green arrows underneath) at 130 kDa upon Western blotting. In contrast, a CFTR-C-band (160 kDa) was seen only in the 16HBE14o-cells. Monocytes and CD4^+^ T cells also displayed undefined signals at ~95 kDa (red arrows underneath). In addition, in all cells except THP-1 and monocytes, another undefined band of ~220 kDa was detected and judged as a non-CFTR protein. It was detected by CFTR antibodies due to incorrect binding, as described previously [19]. Vinculin was used as a loading control.

**Figure 2 ijms-25-06367-f002:**
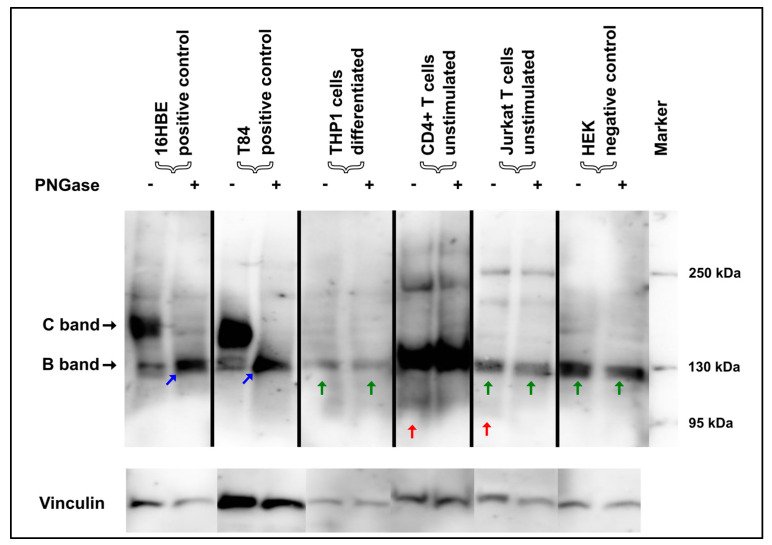
Glycolytic PNGase digest in immune cells. 16HBE14o- and T84 cells showed a typical shift of both the core-glycosylated CFTR-B and the complex-glycosylated CFTR-C towards the CFTR-A (blue arrow), which constitutes the “naked” protein without any glycosyl side chains, but no such shift was seen for any of the immune cells. In contrast, immunoreactive signals compatible with CFTR-B in size in CD4^+^ and Jurkat T cells and THP1 cells did not convert to unglycosylated CFTR-A (green arrow). CD4^+^ and Jurkat T cells displayed non-specific bands at 95 kDa that resolved during PNGase digestion (red arrow).

**Figure 3 ijms-25-06367-f003:**
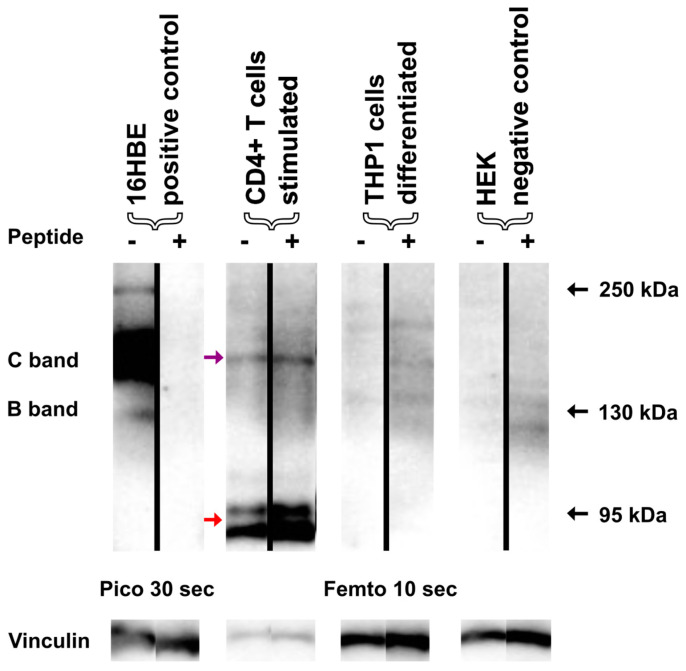
Peptide competition of the CFTR-specific antibody Ab 596. In order to determine specific binding of the CFTR-specific antibody Ab 596 used in the previous experiments, we performed a peptide competition experiment. Lysates of stimulated CD4^+^ T cells and differentiated THP1 cells were immunoblotted with the antibody Ab 596, and either a surplus of the epitope-specific peptide H-WPSGGQMT-OH, PBS only, 16HBE14o-, or HEK cells served as positive and negative controls. As expected, peptide competition completely blocked the CFTR signal (B- and C-band) in 16HBE14o- cells. In contrast, the previously observed bands (red arrows) at 95 kDa in CD4^+^ T cells were not blocked by the peptide. Another immunoreactive narrow band of the size of CFTR-C (purple arrow) was not blocked as well, revealing the non-specific binding of the antibody. Faint signals seen for differentiated THP1 cells were not blocked, and no signals were detected for negative control HEK cells with the AB596. Source data for this figure are provided in Appendix A, which shows the exposures side-by-side for both membranes.

**Table 1 ijms-25-06367-t001:** Antibodies used for immunoblotting. Antibodies against CFTR are raised to target different domains of the complex CFTR protein: ab660 (epitope NBD1, aa 405–436), ab217 (epitope RD, aa 807–819), ab570 (epitope RD, aa 731–742), and ab596 (epitope NBD2, aa 1204–1211).

Antibody	Host/Clonality	Dilution	Provider	Cat. No.
CFTR	Mouse/ Monoclonal	1:400	CF Foundation (Bethesda, MD, USA)	217, 660, 570, 596
Vinculin	Mouse/ Monoclonal	1:500	Abcam (Cambridge, UK)	ab130007
IgG H&L (HRP)	Goat anti-mouse	1:7500	Abcam (Cambridge, UK)	ab97040, Abcam

**Table 2 ijms-25-06367-t002:** CFTR cDNA expression in immune cell lines and primary immune cells was assessed by qPCR and referenced against the expression level of epithelial T84 cells.

Probe **	Amount (ng)	CT Value CFTR	Calibration on T84 (Based on Amount)	δCT (Probe—CALIBRATION) *	Fold Difference T84 ^#^
16HBE	50	20.2			
T84 ^2^	50	20.6		0.0	1
THP-1 undifferentiated ^1^	50	Undetermined	20.6	CFTR mRNA beyond lower limit of detection	
THP-1 differentiated ^1,2,3,S1^	50	34.4	20.6	13.8	670
Jurkat T cells **^,S1^	50	30.6	20.7	9.9	105
CD14^+^ Monocytes ^1^	50	Undetermined	0.0	CFTR mRNA beyond lower limit of detection	
CD19^+^ B cells ^1^	50	Undetermined	20.2	CFTR mRNA beyond lower limit of detection	
CD8^+^ T cells ^1^	50	Undetermined	20.2	CFTR mRNA beyond lower limit of detection	
CD16^+^ NK cells ^1^	50	36.5	20.2	16.2	2067
CD4^+^ T cells unstimulated ^1,2,S1^	50	35.2	20.2	15.0	934
CD4^+^ T cells stimulated ^S3,3^	30	34.9	21.4	13.5	585

** RNA preparation was carried out independently twice. qPCR analyses were run in duplicate. Table displays mean values of all analyses per biomaterial source. * δCT = CT_(T84)_ − CT_(Immune cell)_; ^#^ Fold difference: 1.6 ^δCT^, ^1^ Western blot depicted in Figure 1. ^2^ Protein detection depicted in Figure 2. ^3^ Protein detection depicted in Figure 3. ^S1^ Protein detection depicted in Appendix A. ^S3^ Protein detection depicted in Appendix A.

## Data Availability

Data is contained within the article and Appendix A.

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
