# Peer review of "Analysis of CFTR mRNA and Protein in Peripheral Blood Mononuclear Cells via Quantitative Real-Time PCR and Western Blot"

_ijms, 2024, doi:10.3390/ijms25126367_

Round 1

Reviewer 1 Report

Comments and Suggestions for Authors

In the manuscript by Schnell et al., the authors study a subset of immune cells for their mRNA and protein expression of CFTR. A subset of these cells low amounts of CFTR mRNA were present, however immunoblot analysis failed to detect specific CFTR signals.

Main points:

1.     CFTR function in immune cells is predominantly discussed in the literature for neutrophils, alveolar macrophages and dentritic cells, cell types that are not studied in the current manuscript. This has to be reflected in the manuscript e.g. by changing the title to “some immune cell types” and incorporating this information in the discussion.

2.      The authors use a mixture of 4 verified antibodies with the hope that mixing these antibodies increases the sensitivity (a claim that is not verified in this manuscript). This mixture of antibodies results in some unspecific bands , of which a 130kDa band is particularly problematic since it runs at the same molecular weight as the CFTR band B. It is unlikely that all 4 antibodies give rise to this unspecific signal. Therefore the antibodies have to be tested individually in order to identify single antibodies or combinations that do not result in this unspecific protein detection.

3.      The authors utilize PNGase F digestion to determine the specificity of the CFTR signal. While this method has been widely used to confirm CFTR band C signals, the minute change between CFTR band B and band A cannot be resolved by this technique. This is evident in figure 2 and supplementary figure 2 where the molecular weight of band B signals detected in untreated 16HBE cell lysates overlap with the molecular weight of band A signals after PNGase F treatment. Therefore this method is not adequate to exclude the possibility of band B CFTR presence in immune cells. The peptide inhibition experiments however support this conclusion.

Minor points:

-        Lines 228-231 and 235-238 are repetitive.

-        QPCR results for 16HBE cells are missing from table 1.

-        Are the THP1 cells in figure 2 differentiated or undifferentiated?

-        Figure 3 and others, please add dividing lines to indicate that individual lanes originate from different immunoblots.

-        Figure 1 is shown after figure 2 and 3. Please change the order.

-        The signal for CD4+ T cells in figure 1 looks very different from the signals in figure 2 and 3. Why?

-        In some instances reference 19 and 20 have been mixed up. Please check.

-        A recent manuscript (PMID: 35031577) uses bone marrow transplantation and monocyte-specific CFTR knockout to determine functional role of CFTR expression in this cell type. These results have to be discussed.

Author Response

Main points:

  1. CFTR function in immune cells is predominantly discussed in the literature for neutrophils, alveolar macrophages and dentritic cells, cell types that are not studied in the current manuscript. This has to be reflected in the manuscript e.g. by changing the title to “some immune cell types” and incorporating this information in the discussion.

We thank the reviewer for his suggestion and changed the title of the manuscript into "Detection of CFTR mRNA and Protein in Peripheral Blood Mononuclear Cells via Quantitative Real-Time PCR and Western Blot".

  1. The authors use a mixture of 4 verified antibodies with the hope that mixing these antibodies increases the sensitivity (a claim that is not verified in this manuscript). This mixture of antibodies results in some unspecific bands, of which a 130kDa band is particularly problematic since it runs at the same molecular weight as the CFTR band B. It is unlikely that all 4 antibodies give rise to this unspecific signal. Therefore, the antibodies have to be tested individually in order to identify single antibodies or combinations that do not result in this unspecific protein detection.

We see that the reviewer has some doubts about the efficacy of the antibody mixture used in our study. We would therefore like to explain that the 4 antibodies we obtain from the CFTR Antibody Distribution Program (University of North Carolina at Chapel Hill) bind to different epitopes of the complex structure of the CFTR protein:

660 (epitope NBD1, aa 405-436)

217 (epitope RD, aa 807-819)

570 (epitope RD, aa 731-742)

596 (epitope NBD2, aa 1204-1211).

The respective band patterns of the individual antibodies can be viewed on the website of the CFTR Antibody Distribution Program (https://cftrantibodies.web.unc.edu/western-blot/).

The mixture of antibodies used here is well established and has already been published in one of our recent publications (Stanke et al, Front Pharmacol. 2023;14:1114584. doi: 10.3389/fphar.2023.1114584). Finally, as it can be seen in the blots of the peptide combination, a single use of antibody 596 also produces the so-called fake B band (Figure 3).

  1. The authors utilize PNGase F digestion to determine the specificity of the CFTR signal. While this method has been widely used to confirm CFTR band C signals, the minute change between CFTR band B and band A cannot be resolved by this technique. This is evident in figure 2 and supplementary figure 2 where the molecular weight of band B signals detected in untreated 16HBE cell lysates overlap with the molecular weight of band A signals after PNGase F treatment. Therefore this method is not adequate to exclude the possibility of band B CFTR presence in immune cells. The peptide inhibition experiments however support this conclusion.

We agree with the reviewer that band A and band B are hardly to differentiate by immuno blotting. However, in summary with the peptide competition, this is exactly what can be achieved and this is precisely why we opt for several control experiments to verify CFTR expression.

Minor points:

-Lines 228-231 and 235-238 are repetitive.

We apologize for this mistake, we have now corrected this in the manuscript.

-QPCR results for 16HBE cells are missing from table 1.

The reviewer is right that 16HBE cells are not shown in Table 1. This is because it is a well-known fact that 16HBE express CFTR, therefore no new knowledge is gained by including them to the experiment. Instead, T84 cells are used as reference and positive control for the CFTR expression level in immune cells as a reference, since these cells express CFTR at the highest level known.

Are the THP1 cells in figure 2 differentiated or undifferentiated?

We apologize for this missing information. The cells are differentiated THP1 cells. Based on the suggestion of reviewer #2, we switched Figure 2 and Supplementary Figure 2, because Figure S2 provides immunoblots of an additional T cell line, the Jurkat cells.  THP1 cells are now clearly labelled as differentiated.

Figure 3 and others, please add dividing lines to indicate that individual lanes originate from different immunoblots.

In Figure 3, each lane is derived from the same immunoblot. However, the lanes for each cell line were not run side by side, as you can see in the corresponding Supplementary Figure 5.

Indeed, for Figure 2 (new version) and Supplementary Figure 6, two immunoblots are the source for this figure. Hence, we followed your suggestion and added lines between each lane.

Figure 1 is shown after figure 2 and 3. Please change the order.

We apologize for this mistake, we have corrected that.

The signal for CD4+ T cells in figure 1 looks very different from the signals in figure 2 and 3. Why?

The immunoblots in Figure 1 and for PNGase experiments were obtained using a mix of four different antibodies, whereas for the peptide competition (Fig. 3), only Ab 596 was used.

The different signal pattern in Figure 1 therefore can be explained by an unspecific binding behavior of the other used antibodies compared to Figure 3. In the revised version of the manuscript, Figure 2, you can see a similar prominent signal for CD4 T cells as in Figure 1.

In some instances reference 19 and 20 have been mixed up. Please check.

We corrected the corresponding passages.

A recent manuscript (PMID: 35031577) uses bone marrow transplantation and monocyte-specific CFTR knockout to determine functional role of CFTR expression in this cell type. These results have to be discussed.

Thank you for this important piece of information. We have included a passage citing that reference in the discussion section: “Recently, Fan et al. demonstrated a direct relation between a CFTR defect in monocytes and a clinical phenotype by using wild-type and CF mixed bone marrow chimeras highlighting the functional relevance of CFTR in these cells.“

Reviewer 2 Report

Comments and Suggestions for Authors

In this investigation, they have verified a notably low expression of CFTR mRNA in certain immune cells, with protein levels falling below detectable limits. Nevertheless, the potential for CFTR protein functionality remains plausible. Consequently, confirming the presence of CFTR protein through electrophysiological methods is imperative.

other comments.

1. The precise origin of the 16HBE14o-, THP1, and Jurkat cell lines remains unknown. Detailed information regarding their source needs to be provided and documented.

2. While it is indicated that the primary immune cells were sourced from healthy volunteers, there is an absence of clarity regarding potential ethical considerations. Therefore, it is crucial to ascertain whether the acquisition of these cells underwent scrutiny and approval by the institution's ethics committee, and this information should be provided.

3. Verification is necessary to confirm the accurate quantification of CFTR mRNA through CFTR siRNA knockdown experiments in immune cells, as well as the assessment of CFTR mRNA elevation under experimental conditions stimulating CFTR mRNA expression. There are apprehensions regarding the precise quantification of exceedingly low levels of CFTR mRNA expression.

4. The data presented in Fig. 3 seems to depict discontinuous Western blotting results. To improve clarity, it is essential to explicitly indicate this discontinuity, possibly by adjusting the spacing between lanes. Additionally, it might be more appropriate to reference Fig. S2 instead of Fig. 2, and Fig. S5 instead of Fig. 3 for enhanced accuracy and consistency in the labeling of figures.

5. The location of Fig.1 in the manuscript is inappropriate. There are also noticeable spelling mistakes in methods, etc., so please correct them.

Author Response

Reviewer 2:

In this investigation, they have verified a notably low expression of CFTR mRNA in certain immune cells, with protein levels falling below detectable limits. Nevertheless, the potential for CFTR protein functionality remains plausible. Consequently, confirming the presence of CFTR protein through electrophysiological methods is imperative.

We would like to thank the reviewer for this important commentary. Indeed, McDonald and colleagues have already established CFTR-specific currents lymphocytes as early as in 1992. Furthermore, functional data derived from knock-out models underscore the relevance of an intact CFTR channel in immune cells, although CFTR protein expression in immune cells still remains a debated topic. Therefore, the main focus of our manuscript was to determine CFTR protein expression in these cells which is why we assume that the functional relevance of the CFTR channel has already been demonstrated to a good extent elsewhere.

other comments:

  1. The precise origin of the 16HBE14o-, THP1, and Jurkat cell lines remains unknown. Detailed information regarding their source needs to be provided and documented.

We have included the respective origins of the mentioned cell lines in the method section.  

  • 16HBE14o- were obtained from Prof. Dieter Gruenert (Cell and Genome Engineering Core, University of California, San Francisco)
  • THP-1 (ATCC TIB-202, LGC Standards, Wesel, Germany)
  • Jurkat T cells were obtained from Prof. Hans-Martin Jäck (Division of Immunology, University Hospital Erlangen, Germany)
  1. While it is indicated that the primary immune cells were sourced from healthy volunteers, there is an absence of clarity regarding potential ethical considerations. Therefore, it is crucial to ascertain whether the acquisition of these cells underwent scrutiny and approval by the institution's ethics committee, and this information should be provided.

We agree with the reviewer that this information is crucial. We now provided all relevant information to this point in the revised manuscript: “PBMCs of two healthy, anonymous volunteers were obtained from the Department of Transfusion Medicine and Hemostasis (University Hospital Erlangen, Germany) as a by-product of the generation of erythrocyte concentrates and stored in liquid nitrogen. Experiments with PBMCs derived from patients and probands were approved by the local ethics committee (vote # 20-485_1-B).” 

  1. Verification is necessary to confirm the accurate quantification of CFTR mRNA through CFTR siRNA knockdown experiments in immune cells, as well as the assessment of CFTR mRNA elevation under experimental conditions stimulating CFTR mRNA expression. There are apprehensions regarding the precise quantification of exceedingly low levels of CFTR mRNA expression.

CFTR mRNA expression in immune cells has already been established by Yoshimura / Crystal, 1991. However, the main purpose of our manuscript was to test for CFTR protein expression by Western blot and – as the observed signals turned out to be unspecific – to highlight potential pitfalls one might come across when performing CFTR immunoblots / - fluorescence experiments in immune cells. Furthermore, we would like to point out, that we already have provided data for CFTR mRNA expression in stimulated and unstimulated T cells, as you can read in Table 2.

  1. The data presented in Fig. 3 seems to depict discontinuous Western blotting results. To improve clarity, it is essential to explicitly indicate this discontinuity, possibly by adjusting the spacing between lanes. Additionally, it might be more appropriate to reference Fig. S2 instead of Fig. 2, and Fig. S5 instead of Fig. 3 for enhanced accuracy and consistency in the labeling of figures.

We would like to thank the reviewer for his helpful suggestion to replace Figure 2 with Supplementary Figure 2. However, we decided to keep Figure 3 and Supplementary Figure 5 in its original sequence, as we believe that this order provides the most consistency in terms of the provided information. To improve the structure of Figure 3, we have added some spacing between the lanes as suggested.

  1. The location of Fig.1 in the manuscript is inappropriate. There are also noticeable spelling mistakes in methods, etc., so please correct them.

We have corrected this in the revised version of the manuscript.

Reviewer 3 Report

Comments and Suggestions for Authors

The current manuscript attempts to address an important question with regards to CFTR expression in non-epithelial cells - here there is currently a lot of conflicting data. A throrough investigation of CFTR mRNA and protein expression in immune cells is a much needed step forward in understanding how CF is affecting these cells.

Major comments:

L126: Given the variability of immune cells between healthy volunteers, are two healthy volunteers sufficient for this study?

L243: Despite mentioning 16HBE as positive control mRNA expression for this cell line is not given and should be added to the table

L310: In Figure 1, can the authors specify how they discriminate between the false WB bands at ~130 and real signal, given that this band is also present in the negative control HEK cells

In general, how did the authors control for the large differences in expression levels of CFTR mRNA/Protein in epithelial vs immune cells?

In the discussion, could the authors elaborate on the available data of CFTR modulators on the cells studied in this manuscript?

Minor comments:

L59: As neutrophils (or HL60 cell line) are not included in the study, could the authors elaborate on what is known about CFTR expression/function in these cells

L165/166: There appears to be some duplication of the cell line names, please check

L195: Table 2 appears before Table 1 & Figure 1 was near the end of the manuscript & in general the numbers should be checked

Table 1: Stimulated CD4+ cells are mentioned to be in Figure 1 but this is not the case & T84 cells are mentioned to be in S1 but this is also not the case

Comments on the Quality of English Language

English is fine - minor spellcheck recommended.

Author Response

The current manuscript attempts to address an important question with regards to CFTR expression in non-epithelial cells - here there is currently a lot of conflicting data. A throrough investigation of CFTR mRNA and protein expression in immune cells is a much needed step forward in understanding how CF is affecting these cells.

Major comments:

L126: Given the variability of immune cells between healthy volunteers, are two healthy volunteers sufficient for this study?

In general, we agree with the reviewer that n=2 lacks the power to detect statistically significant differences. However, the purpose of our study was not to determine interindividual differences regarding CFTR expression but to establish appropriate quality controls for CFTR qPCR and Western blot. In that context, we would think that n=2 is sufficient for that specific purpose.

L243: Despite mentioning 16HBE as positive control mRNA expression for this cell line is not given and should be added to the table.

We did not perform qPCR experiment with 16HBE. Here, the contradictory passage was corrected in the revised version of the manuscript.

L310: In Figure 1, can the authors specify how they discriminate between the false WB bands at ~130 and real signal, given that this band is also present in the negative control HEK cells

To that account, we have performed experiments using PNGase digestion and peptide competition as described in the manuscript.

In general, how did the authors control for the large differences in expression levels of CFTR mRNA/Protein in epithelial vs immune cells?

Regarding the qPCR experiments, the amount of template RNA was normed to 50 or 30 ng. The CT-values for CFTR cDNA were than adjusted to the expression level of T84 cells as reference sample.

For the immunoblots, also the same amount of each cell lysate was used. Futhermore, Vinculin staining served as quality control for membrane loading. In general, densitometry of the CFTR band would also have been performed but was omitted since no CFTR signal was detected in the cells of interest.

In the discussion, could the authors elaborate on the available data of CFTR modulators on the cells studied in this manuscript?

We have included a section on the effect of CFTR modulator therapies on immune cells.

Minor comments:

L59: As neutrophils (or HL60 cell line) are not included in the study, could the authors elaborate on what is known about CFTR expression/function in these cells

Certainly, neutrophils as main drivers of lung inflammation in CF would also be of particular interest regarding CFTR expression and function. However, we specifically decided to omit them from the study, as polymorphonuclear cells cannot be stored in liquid nitrogen - in contrast to PBMCs. As they are not part of the study, we think that discussion CFTR expression and functions would be out of focus of a concise discussion of the relevant results.

L165/166: There appears to be some duplication of the cell line names, please check

We have corrected this.

L195: Table 2 appears before Table 1 & Figure 1 was near the end of the manuscript & in general the numbers should be checked

We have corrected this.

Table 1: Stimulated CD4+ cells are mentioned to be in Figure 1 but this is not the case & T84 cells are mentioned to be in S1 but this is also not the case

We have corrected this.

Round 2

Reviewer 1 Report

Comments and Suggestions for Authors

In the first round of revision the authors did not adequately address some of the issues raised:

1. Regarding the mixture of 4 antibodies for the detection of CFTR: I agree with the authors that it is likely that mixing antibodies recognizing different epitopes increases the detection efficacy. I also agree that the CFTR Antibody Distribution Program is a very reputable source for specific CFTR antibodies. However, in this manuscript an unspecific band at a similar MW as the CFTR band B is detected. Since the antibodies were raised in different animals (each with a unique immune system), it is very unlikely that all 4 antibodies detect the same unspecific protein. Therefore, as I asked before, the antibodies have to be tested individually in order to identify single antibodies or combinations that do not result in this unspecific protein detection. The authors argue “as it can be seen in the blots of the peptide combination, a single use of antibody 596 also produces the so-called fake B band (Figure 3)”. This may suggest that the antibody mixture lacking the 596 antibody does not detect the unspecific protein at ~130kDa. This has to be tested.

2. In the previous review I pointed out to the authors that band A and B cannot be resolved by immunoblotting. The authors agree with this “We agree with the reviewer that band A and band B are hardly to differentiate by immuno blotting”. Please change the manuscript text in section 3.3. to reflect this.

Author Response

Reviewer #1:

In the first round of revision the authors did not adequately address some of the issues raised:

  1. Regarding the mixture of 4 antibodies for the detection of CFTR: I agree with the authors that it is likely that mixing antibodies recognizing different epitopes increases the detection efficacy. I also agree that the CFTR Antibody Distribution Program is a very reputable source for specific CFTR antibodies. However, in this manuscript an unspecific band at a similar MW as the CFTR band B is detected. Since the antibodies were raised in different animals (each with a unique immune system), it is very unlikely that all 4 antibodies detect the same unspecific protein. Therefore, as I asked before, the antibodies have to be tested individually in order to identify single antibodies or combinations that do not result in this unspecific protein detection. The authors argue “as it can be seen in the blots of the peptide combination, a single use of antibody 596 also produces the so-called fake B band (Figure 3)”. This may suggest that the antibody mixture lacking the 596 antibody does not detect the unspecific protein at ~130kDa. This has to be tested.

We thank the reviewer for this important point. We have performed an additional experiment as suggested testing each of the four used antibodies separately. This experiment reveals Ab660 as origin of the strong unspecific signal in the immunoblot in THP1 and CFTR-negative HEK cells. In turn, the other 3 antibodies (217, 570 and 596) do not produce this unspecific signal and fail to raise a CFTR-specific signal in THP1 and HEK cells as their specific CFTR epitopes are not recognized.

Taken together with our findings of glycolytic digestion and peptide competition, we believe that our results do not support the hypothesis of a CFTR protein expression in PBMCs.

  1. In the previous review I pointed out to the authors that band A and B cannot be resolved by immunoblotting. The authors agree with this “We agree with the reviewer that band A and band B are hardly to differentiate by immuno blotting”. Please change the manuscript text in section 3.3. to reflect this.

We have corrected this point in the manuscript and emphasized that mainly the shift of C- to B-band following glycolytic digest can be observed: „[A]fter the enzymatic digest with PNGase, the complex-glycosylated C-bands of the CFTR protein shifted towards the core-glycosylated B-band and an additional a band s

Reviewer 2 Report

Comments and Suggestions for Authors

The author has appropriately addressed the issues requested. All discontinuous western blot data should be clearly marked as discontinuous (Fig.3). I think that point should be corrected.

Author Response

Reviewer #2:

The author has appropriately addressed the issues requested. All discontinuous western blot data should be clearly marked as discontinuous (Fig.3). I think that point should be corrected.

We have corrected this in the manuscript and have added additional vertical lines in order to highlight the discontinuous character of the blots.

Reviewer 3 Report

Comments and Suggestions for Authors

I thank the authors for this revised version of the manuscript and for clarifying certain aspects, although I think the discussion on CFTR modulator treatments (and the proposed off target effects) could be a bit more in depth.

However, I still believe that 16HBE qPCR data should be added to the manuscript since this is the control used in the remainder of the manuscript.

Comments on the Quality of English Language

English is fine, minor spellcheck recommend. Some obvious spelling errors have been corrected

Author Response

I thank the authors for this revised version of the manuscript and for clarifying certain aspects, although I think the discussion on CFTR modulator treatments (and the proposed off target effects) could be a bit more in depth.

We thank reviewer #3 for his comments and critical thoughts in order to improve our paper. With regard to CFTR modulators, we have opted to not extensively discuss those in the current manual as the main focus of our work here is a methodological one regarding the CFTR expression in immune cells derived from healthy controls. That’s why we feel that a discussion on CFTR modulators would rather confuse the reader than help him to better understand our findings.

However, I still believe that 16HBE qPCR data should be added to the manuscript since this is the control used in the remainder of the manuscript.

As suggested, we have included qPCR data for CFTR expression in 16HBE cells that are in the same range as T84.

Round 3

Reviewer 1 Report

Comments and Suggestions for Authors

The manuscript has been substantially improved and all my queries were answered. I don't have further concerns and recommend to accept the manuscript.